# Neural-Symbolic Multi-Objective Optimization for Performance-Aware ORM Database Design

Sasan Azizian
College of Science and Technology,
Bellevue University
Bellevue, Nebraska, USA
sazizian@bellevue.edu

Ayoub Hazrati
The Vanguard Group
Valley Forge, Pennsylvania, USA
hazrati.ayoub@gmail.com

Artin Azizian
School of Computer Science, McGill
University
Montreal, Quebec, Canada
artin.azizian@mail.mcgill.ca

Elham Rastegari
Business Intelligence & Analytics,
Creighton University
Omaha, Nebraska, USA
elhamrastegari@creighton.edu

Hamid Bagheri
School of Computing, University of
Nebraska–Lincoln
Lincoln, Nebraska, USA
bagheri@unl.edu

Juan Cui
School of Computing, University of
Nebraska–Lincoln
Lincoln, Nebraska, USA
jcui@unl.edu

## ABSTRACT

Object–relational mapping (ORM) design remains largely driven by fixed heuristics that fail to capture workload-specific tradeoffs among query latency, insert cost, and memory footprint. We present Y-Map, a hybrid neural–symbolic framework for performance-aware ORM schema design that synthesizes *valid* schema candidates and predicts their performance without requiring workload execution at inference time. Y-Map leverages Alloy to enumerate correctness-preserving ORM schemas and ranks them using a multi-encoder regression model that fuses structural, syntactic, and semantic representations with compact schema-level features. By predicting continuous performance objectives, accounting for latency, query latency, and memory footprint, Y-Map enables Pareto-aware selection without per-candidate benchmarking during inference. We evaluate Y-Map on nine object models from e-commerce, banking, and healthcare. Relative to two representative baselines, Leant and DTS, Y-Map yields improved aggregate Pareto quality (GD/HV) while reducing inference time and memory overhead. Overall, the results show that integrating symbolic validity guarantees with learned performance prediction provides a practical, scalable solution for workload-aware ORM optimization.

## CCS CONCEPTS

• **Software and its engineering** → **Formal software verification**.

## KEYWORDS

Object–Relational Mapping, Neural–Symbolic Optimization, Performance Prediction, Multi-objective Optimization, Schema Design

**ACM Reference Format:**
Sasan Azizian, Ayoub Hazrati, Artin Azizian, Elham Rastegari, Hamid Bagheri, and Juan Cui. 2026. Neural-Symbolic Multi-Objective Optimization for Performance-Aware ORM Database Design. In *Proceedings of 3rd ACM International Conference on AI-powered Software (AIware '26)*. ACM, New York, NY, USA, 5 pages. https://doi.org/XXXXXXX.XXXXXXX

## 1 INTRODUCTION

Relational databases remain the dominant foundation for persistent data management, while object-oriented programming structures application logic. Object–relational mapping (ORM) frameworks such as Hibernate, Django ORM, SQLAlchemy, and Entity Framework bridge this mismatch by translating object models into relational schemas and SQL operations [1–3, 11, 12, 18]. However, ORM *design choices*—including inheritance strategy, association encoding, and normalization level—can substantially affect performance by altering join depth, redundancy, index opportunities, and update behavior [3, 10, 12, 17]. A single object model often admits many correctness-preserving mappings, each inducing different tradeoffs among query latency, insert cost, and memory footprint. In practice, mainstream ORMs rely largely on fixed defaults and heuristics, leaving developers to diagnose and revise mappings late in the development process. Prior work has improved ORM design along three main directions. *Symbolic synthesis* approaches (e.g., *TradeMaker*) encode mapping rules in relational logic and use Alloy to enumerate valid schemas, providing correctness by construction but facing combinatorial growth as models scale [9, 16]. *Benchmarking-driven* approaches (e.g., Leant) associate synthesized candidates with measured performance, but incur substantial cost for schema deployment, data loading, and workload execution [8]. *Learning-based* methods (e.g., DTS) improve scalability by identifying near-Pareto candidates, yet coarse labels can obscure fine-grained tradeoffs and limit interpretability [14]. As a result, a gap remains between methods that guarantee validity but are expensive, and methods that scale but do not provide continuous, interpretable performance estimates. We present Y-Map, a hybrid neural–symbolic framework for *benchmark-free inference* in multi-objective ORM optimization. Y-Map uses Alloy to synthesize correctness-preserving candidates under application constraints and then predicts continuous performance objectives—insert latency, query latency, and memory footprint—without executing the workload at inference time. To

capture schema behavior across heterogeneous domains, Y-MAP combines compact symbolic features (e.g., foreign-key count, join-table usage, inheritance depth, and normalization indicators) with a multi-encoder representation that fuses structural embeddings from GraphCodeBERT, syntactic embeddings from CodeT5+, and semantic embeddings from LLaMA [13, 19, 20]. A regression head predicts a continuous performance vector, enabling Pareto-aware selection and evaluation using standard multi-objective metrics such as GD and HV [15, 21].We evaluate Y-MAP on nine object models spanning e-commerce, banking, and healthcare. Relative to representative benchmarking-driven and learning-based baselines, Y-MAP improves aggregate Pareto quality while reducing inference-time cost [8, 14]. A related AIware 2026 paper, TRIORM, investigates workload-aware neural–symbolic ORM optimization [6]. A preliminary version of this work appeared at ICSE 2026 [7].

**Contributions. (1) Neural–symbolic ORM optimization:** a framework that synthesizes correctness-preserving ORM schemas and ranks them using predicted multi-objective performance, eliminating workload execution at inference time. **(2) Multi-encoder regression:** a fused schema representation combining GraphCode-BERT, CodeT5+, and LLaMA embeddings with symbolic features to predict continuous insert latency, query latency, and memory footprint. **(3) Pareto-aware selection:** an inference procedure that supports both predicted Pareto frontier extraction and scalarized schema selection under user-specified objective preferences.

## 2 PROBLEM DEFINITION AND OVERVIEW

**Object model and constraints.** We represent an application domain as a typed graph $O = (C, \mathcal{R}, \mathcal{H})$, where $C$ denotes classes with attributes, $\mathcal{R}$ denotes associations with multiplicities, and $\mathcal{H} \subseteq C \times C$ denotes inheritance relations [3, 12, 17]. Let $\Phi$ denote admissibility constraints over ORM realizations, including allowed inheritance strategies (e.g., single-table, joined-subclass, table-per-class), association encodings (foreign key versus join table), and relational integrity requirements such as keys, referential integrity, and multiplicity preservation [10, 12, 17]. Let $W$ denote a workload profile describing the insert/query mix. **Valid candidates.** Each mapping induces a relational schema $S = (\mathcal{T}, \mathcal{K}, \mathcal{F})$, where $\mathcal{T}$ denotes tables and columns, $\mathcal{K}$ keys, and $\mathcal{F}$ foreign keys. A schema is *valid* if it preserves object-model semantics while satisfying $\Phi$:

$$S \models (O, \Phi), \qquad \mathcal{S}(O, \Phi) = \{ S \mid S \models (O, \Phi) \}.$$

**Objectives and goals.** Each valid schema induces a latent performance vector under $W$,

$$Y(S) = \big( T_{\text{ins}}(S), T_{\text{qry}}(S), M(S) \big),$$

where the objectives correspond to insert latency, query latency, and memory footprint. These costs are strongly influenced by join structure and redundancy [3, 12]. Our goal is to identify the Pareto-efficient set

$$\mathcal{P}^{\star} = \{ S \in \mathcal{S}(O, \Phi) \mid \nexists S' \in \mathcal{S}(O, \Phi) : Y(S') \prec Y(S) \},$$

and evaluate tradeoff quality using standard multi-objective metrics such as GD and HV [15, 21]. **Challenge.** The candidate space $|\mathcal{S}(O, \Phi)|$ grows combinatorially with inheritance and association choices, while obtaining $Y(S)$ through schema deployment and workload execution is expensive at scale [8, 9]. Y-MAP addresses

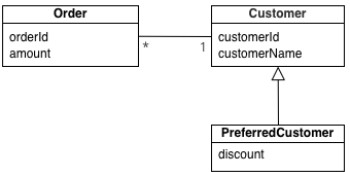

**Figure 1: Object model (`Customer–Order`) with inheritance and a 1:N association.**

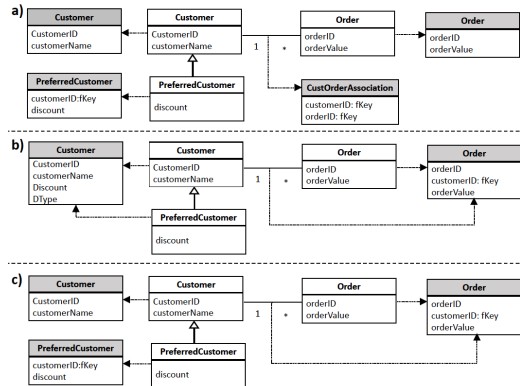

**Figure 2: Representative valid ORM mappings for Figure 1. Different inheritance and association choices alter join structure, redundancy, and constraint enforcement.**

this challenge by preserving validity through synthesis and replacing per-candidate benchmarking at inference time with learned performance prediction for tradeoff-aware ranking.

## 3 ILLUSTRATIVE EXAMPLE

Even small object models can yield a large ORM design space: mappings that are semantically equivalent at the application level may produce different relational schemas and performance tradeoffs. Figure 1 shows a simple `Customer–Order` model with inheritance (`PreferredCustomer → Customer`) and a 1:N association to `Order`. Under standard ORM mapping rules and integrity constraints, the model admits multiple correctness-preserving relational realizations [3, 12, 17]. Figure 2 presents representative valid mappings that vary in inheritance strategy and association encoding, resulting in different join structures, redundancy, and constraint patterns.

**Tradeoff intuition.** Normalized designs typically reduce redundancy but increase join cost, while flatter designs reduce joins at the expense of duplicated attributes, higher insert overhead, and larger storage footprint [3, 10, 12]. Association encodings further shift these tradeoffs. This motivates predicting continuous objectives ($T_{\text{ins}}, T_{\text{qry}}, M$) to rank *valid* mappings without per-candidate benchmarking.

## 4 APPROACH

Y-MAP is a neural–symbolic framework for selecting *performance-efficient* yet *valid* ORM mappings. It decouples (i) *offline labeling and model training* from (ii) *inference-time* optimization, where candidate schemas are ranked *without* per-candidate deployment, data loading, or workload execution. Figure 3 overviews the pipeline.

### 4.1 Validity-Preserving Candidate Generation

Given an object model $O$ and mapping constraints $\Phi$ (e.g., permissible inheritance and association strategies, multiplicity preservation,

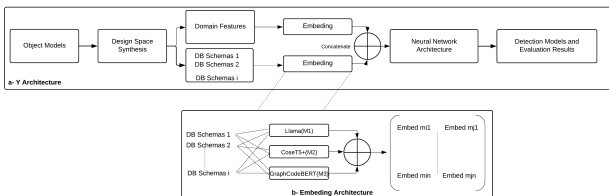

**Figure 3: Y-Map pipeline. The framework synthesizes valid ORM schemas from $(O, \Phi)$, constructs fused neural–symbolic representations, and predicts performance objectives for Pareto-aware selection.**

and key/foreign-key constraints), Y-Map constructs an Alloy specification $\mathcal{A} = \text{Trans}(O, \Phi)$ and enumerates a bounded set of candidate schemas

$$\mathcal{S} = \text{Synth}(\mathcal{A}) = \{S_1, \ldots, S_n\}, \qquad S_i \models (O, \Phi), \qquad (1)$$

using the Alloy Analyzer [16]. This synthesis step follows the specification-driven ORM literature and guarantees that every $S_i$ satisfies the encoded mapping rules by construction [9]. For each candidate $S_i$, we also compute a compact symbolic feature vector $F_i \in \mathbb{R}^q$, capturing lightweight structural statistics that are predictive of ORM performance (e.g., #tables, #foreign keys, join-table usage, inheritance depth, and coarse normalization indicators).

## 4.2 Multi-Encoder Schema Representation

Each schema $S_i$ is serialized into a canonical SQL DDL sequence $X_i$ with deterministic ordering and identifier normalization (e.g., consistent renaming of tables/columns), reducing spurious lexical variation across domains. We then compute a multi-view embedding using three pretrained encoders that capture complementary information: GraphCodeBERT models structural dependencies (e.g., relationships induced by key/foreign-key edges) [13], CodeT5+ captures syntactic regularities and schema idioms [20], and LLaMA contributes broader contextual semantics for robustness across heterogeneous schemas [19]. For encoder $m \in \{1, 2, 3\}$,

$$E_i^{(m)} = f_m(X_i), \qquad E_i = \text{concat}\left(E_i^{(1)}, E_i^{(2)}, E_i^{(3)}\right). \qquad (2)$$

Finally, neural and symbolic views are fused into a unified representation:

$$Z_i = [E_i \| F_i]. \qquad (3)$$

We **freeze** the pretrained encoders and train only a lightweight regression head (a $k$-layer MLP with ReLU) on the offline-labeled schemas[4, 5].

## 4.3 Continuous Performance Regression

Y-Map learns a regression model $g_\theta$ that predicts a continuous 3-objective performance vector for each schema:

$$\hat{Y}_i = g_\theta(Z_i) = \left(\hat{T}_{\text{ins}}(S_i), \hat{T}_{\text{qry}}(S_i), \hat{M}(S_i)\right), \qquad (4)$$

where $\hat{T}_{\text{ins}}$ and $\hat{T}_{\text{qry}}$ denote predicted insert and query latency, and $\hat{M}$ denotes predicted memory footprint. Training uses an *offline-labeled* dataset of schema–performance pairs. Ground-truth labels are obtained by materializing each $S_i$ and profiling a controlled workload in a fixed DBMS configuration, consistent with benchmark-driven ORM optimization pipelines [8]. The model is trained by minimizing mean squared error:

$$\mathcal{L}(\theta) = \frac{1}{N} \sum_{i=1}^{N} \|g_\theta(Z_i) - Y_i\|_2^2. \qquad (5)$$

**Algorithm 1** Inference-time ranking in Y-Map

---
**Require:** $O, \Phi, g_\theta$, optional $(\alpha, \beta, \gamma)$
0:   $A \leftarrow \text{Trans}(O, \Phi)$
0:   $\mathcal{S} \leftarrow \text{Synth}(A) \{\forall S \in \mathcal{S} : S \models (O, \Phi)\}$
0:   **for all** $S_i \in \mathcal{S}$ **do**
0:     $F_i \leftarrow \text{Feat}(S_i); X_i \leftarrow \text{DDL}(S_i)$
0:     $E_i \leftarrow \text{concat}(f_1(X_i), f_2(X_i), f_3(X_i))$
0:     $\hat{Y}_i \leftarrow g_\theta([E_i \| F_i])$
0:   **end for**
0:   **if** $(\alpha, \beta, \gamma)$ is provided **then**
0:     **return** $\arg\min_{S_i \in \mathcal{S}} \alpha \hat{T}_{\text{ins}}(S_i) + \beta \hat{T}_{\text{qry}}(S_i) + \gamma \hat{M}(S_i)$
0:   **else**
0:     **return** $\text{Pareto}(\{\hat{Y}_i\})$
0:   **end if**=0
---

**Inference-time behavior.** At inference time, Y-Map avoids per-candidate workload execution: it only computes $(F_i, X_i)$, encodes $X_i$, and predicts $\hat{Y}_i$. This shifts the expensive benchmarking cost to an offline labeling stage, enabling fast ranking over large candidate sets.

## 4.4 Pareto-Aware Optimization and Selection

Given predicted vectors $\{\hat{Y}_i\}$, Y-Map supports two standard decision modes. First, it can return the predicted Pareto set

$$\text{Pareto}(\{\hat{Y}_i\}) = \{S_i \in \mathcal{S} \mid \nexists S_j : \hat{Y}_j \preceq \hat{Y}_i \wedge \hat{Y}_j \neq \hat{Y}_i\},$$

providing developers with explicit tradeoffs among insert latency, query latency, and memory usage. Second, when users provide objective weights, Y-Map selects a single schema via scalarization:

$$S^\star = \arg\min_{S_i \in \mathcal{S}} \alpha \hat{T}_{\text{ins}}(S_i) + \beta \hat{T}_{\text{qry}}(S_i) + \gamma \hat{M}(S_i), \qquad (6)$$

with $\alpha, \beta, \gamma \geq 0$. Compared to Pareto-membership classifiers, continuous regression enables fine-grained ranking *within* the frontier and across near-frontier candidates, supporting more informative design-time decision-making [14]. **End-to-End Procedure** Algorithm 1 summarizes inference: Y-Map synthesizes valid candidates, constructs a fused representation per candidate, predicts continuous objectives, and finally performs Pareto filtering or scalarized selection.

## 5 EVALUATION

We evaluate Y-Map along two axes: **(i) tradeoff quality**—how well its predicted non-dominated set approximates the ground-truth Pareto frontier over insert latency, query latency, and memory footprint—and **(ii) efficiency**—end-to-end inference time and peak memory relative to prior symbolic+benchmarking and learning-based baselines. Importantly, Y-Map performs *no per-candidate* deployment, data loading, or workload execution at inference time; all runtime measurements reported for Y-Map correspond to representation construction, prediction, and Pareto filtering.

### 5.1 Experimental Setup

**Subjects and candidate generation.** We use nine object models spanning e-commerce, banking, and healthcare, covering both small benchmark schemas and two enterprise-scale models to stress scalability. For each model, we compile $(O, \Phi)$ into an Alloy specification and synthesize a bounded candidate set $\mathcal{S}$ using the Alloy Analyzer [16]. By construction, every candidate $S \in \mathcal{S}$ satisfies the mapping constraints (i.e., $S \models (O, \Phi)$), following the specification-driven ORM synthesis methodology established by prior work [9]. Across the 9 models, Alloy synthesis produced **100–2700** valid

schemas per model; we obtained ground-truth labels for approximately **9,500** schemas via offline profiling (across nine benchmark object models). **Offline labeling (ground truth) and workload.** To obtain training/evaluation labels, each synthesized schema is materialized as SQL DDL in PostgreSQL and profiled *once offline* using a controlled workload consisting of bulk inserts and representative SELECT templates. We record insert latency and query latency (ms) and memory footprint measured as database size (MB), consistent with benchmark-driven ORM profiling pipelines [8]. These offline measurements are used only to train the regressor and to compute evaluation references; at inference time, Y-Map ranks candidates using predicted metrics without executing the workload. **Baselines.** To keep the short-paper evaluation focused, we compare against two representative and strong baselines: Leant, a symbolic+benchmarking approach that learns from profiled candidates [8], and DTS (DesignTradeoffSculptor), a transformer-based approach targeting Pareto-relevant learning [14]. All methods operate on identical candidate pools produced under matched Alloy bounds. **Protocol and metrics.** We report multi-objective quality using *Generational Distance (GD)* (lower is better) [15] and *Hypervolume (HV)* (higher is better) [21]. For Y-Map, we additionally report regression fidelity (MAE/RMSE) over the three objectives to validate that Pareto improvements stem from accurate *continuous* predictions rather than coarse near-frontier labeling. Finally, we summarize the end-to-end inference time and peak memory usage. The **reference Pareto frontier** is computed from the same candidate pool using the offline ground-truth measurements and is used only for evaluation. Experiments were conducted on a workstation with a single NVIDIA A100 GPU and using PostgreSQL 14.

## 5.2 Results (Aggregate Summary)

Table 1 summarizes aggregate Pareto quality across the nine systems. Y-Map achieves the lowest GD and the highest HV, indicating that its predicted non-dominated set more closely matches the ground-truth Pareto frontier and covers a larger dominated region than Leant and DTS [8, 14]. Table 3 reports the corresponding efficiency results, showing that Y-Map substantially reduces both inference time and peak memory usage because it requires only representation construction and prediction at inference time, without per-candidate workload execution. **Continuous prediction fidelity.** To show that these Pareto improvements are supported by accurate numerical estimation, Table 2 reports MAE/RMSE on held-out folds for insert latency, query latency, and memory footprint. The consistently low error across all three objectives indicates that Y-Map's gains stem from fine-grained continuous regression rather than coarse Pareto-membership decisions. In contrast to classification- or ranking-based approaches, continuous prediction enables more precise differentiation among near-frontier schemas, leading to improved hypervolume coverage and more interpretable tradeoff analysis. **Takeaway.** Overall, the aggregate results indicate that Y-Map more accurately approximates the ground-truth Pareto frontier, maintains low prediction error, and reduces inference cost relative to strong baselines [8, 14]. Per-system analyses and additional ablations are deferred to the extended version.

## 6 DISCUSSION & LIMITATIONS

**Neural–symbolic design.** Y-Map separates semantic validity from performance reasoning. Validity is guaranteed through Alloy-based

**Table 1: Aggregate Pareto quality over 9 systems (mean over folds). Lower GD and higher HV are better.**

| Method | GD ↓ | HV ↑ |
|---|---|---|
| Leant [8] | 0.07 | 0.61 |
| DTS [14] | 0.02 | 0.85 |
| **Y-Map** | **0.01** | **0.91** |

**Table 2: Continuous prediction accuracy (5-fold CV). MAE/RMSE for the three objectives (lower is better).**

| Metric | MAE | RMSE |
|---|---|---|
| Insert latency (ms) | $5.8 \pm 0.7$ | $6.4 \pm 0.9$ |
| Query latency (ms) | $6.3 \pm 0.8$ | $7.5 \pm 1.0$ |
| Memory (MB) | $7.1 \pm 0.9$ | $7.9 \pm 1.1$ |

**Table 3: Efficiency over 9 systems (mean). Time includes representation, prediction, and Pareto filtering.**

| Method | Time (s) ↓ | Peak Mem. (GB) ↓ |
|---|---|---|
| Leant [8] | 26,000 | 17.0 |
| DTS [14] | 3,400 | 9.0 |
| **Y-Map** | **900** | **6.2** |

synthesis, while performance is modeled by a learned regressor over schema representations. This design combines the correctness guarantees of symbolic synthesis with the scalability of learned prediction, addressing the limitations of both benchmarking-heavy and coarse classification-based approaches [9, 14, 16]. **Clarifying "benchmark-free."** In Y-Map, *benchmark-free* refers specifically to inference time: candidate schemas are ranked without per-candidate deployment, data loading, or workload execution. An offline profiling stage is still required for supervision and evaluation, consistent with prior benchmark-driven ORM pipelines [8]. Thus, Y-Map removes benchmarking from the online decision loop rather than eliminating measurement altogether. **Tradeoff reasoning.** By predicting a continuous objective vector ($T_{ins}$, $T_{qry}$, $M$), Y-Map supports both Pareto filtering and preference-aware scalarization. This enables finer-grained tradeoff analysis than approaches based only on Pareto membership or coarse ranking labels [14].

**Limitations.** The approach still relies on offline labeling, and predictive accuracy may degrade when the workload shifts substantially. Reported results are tied to a fixed DBMS configuration, while Alloy-based synthesis remains bounded by analysis scope [16]. In addition, this short paper reports aggregate results against two representative baselines; more detailed per-system analyses are left to future work.

## 7 CONCLUSION

We introduced Y-Map, a neural–symbolic framework for multi-objective ORM schema design. Y-Map combines Alloy-based synthesis with continuous performance prediction to generate *valid* ORM schemas and rank them efficiently without per-candidate workload execution at inference time. Evaluation on nine object models shows that Y-Map improves aggregate Pareto quality (GD/HV), maintains low prediction error, and reduces inference cost compared to Leant and DTS [8, 14]. Future work will broaden the empirical study with per-system analyses, additional baselines, and sensitivity to workload and DBMS variation.

## ACKNOWLEDGMENT

The authors gratefully acknowledge the support of Bellevue University for this research.

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
