# OpenReview forum: "Neural-Symbolic Multi-Objective Optimization for Performance-Aware ORM Database Design"
_ACM.org/AIWare/2026/Conference — AIware 2026_

### Official Review · Reviewer_vaVz · 2026-02-25

**Rating:** 2
**Confidence:** 3

**Review:**

### **Strengths**





- Tackles a practical and relevant problem: ORM mapping choices create real latency/memory tradeoffs, and exhaustive benchmarking is expensive.
- Clean pipeline: symbolic synthesis to ensure validity, followed by learned ranking to reduce inference-time cost; the multi-objective framing (Pareto-based selection) is appropriate for the setting.
- Clear articulation of the intended user workflow (rank candidates, optionally scalarize by preferences).







### **Major Comments**





1. **LLM motivation is unclear and currently feels bolted-on.**

   The paper positions multi-encoder/LLM embeddings (incl. LLaMA) as a key ingredient, but it does not provide a convincing mechanism-level argument for *why* LLM-style semantic representations should improve prediction of system-level metrics (insert/query latency, memory) that are largely driven by structural properties (join patterns, redundancy/normalization, FK/index opportunities). As written, the “LLM → finer-grained tradeoffs” narrative is not sufficiently supported.

2. **Model choice appears arbitrary given the claimed role of the learned representation.**

   If LLaMA is included for robustness/semantics, the paper should justify why a general-purpose LLM is appropriate for SQL DDL / ORM schemas, and why this particular set of encoders (GraphCodeBERT, CodeT5+, LLaMA) is the right design point. In 2026 especially, readers will expect at least a brief rationale (or minimal sensitivity discussion) explaining why these models, and what complementary signal each provides beyond symbolic features.

3. **Positioning relative to prior transformer-based ORM tradeoff work should be sharpened.**

   The overall shape (candidate enumeration + ML/transformer-assisted tradeoff selection) is close to recent work such as [10]. The paper would benefit from an explicit side-by-side comparison of what is done offline vs. online, and what is fundamentally new beyond “no workload execution at inference time,” to avoid the impression of incremental re-packaging with heavier model stacking.



### **Minor Comments**


- **Clarify GD/HV computation details** (normalization and HV reference point), and consider adding a per-system breakdown or simple visualization for interpretability
- No related work. a brief expansion to situate the paper within (i) symbolic ORM synthesis and (ii) learning-based performance/cost modeling in DB/SE would improve readability and reduce the “single-thread” framing.
- Figures are extremely tiny and hard to recognize
- The word "benchmark-free" sounds weird. It could be misread as “no measurements at all.”

[Update]
Thank you for the clarification. It helps narrow the paper’s claims.

However, my main concern remains. The multi-encoder design is still not convincingly motivated: it is still unclear why this particular combination of models is the right design choice, what concrete complementary signal each encoder contributes, and why these signals are needed beyond symbolic features or simpler representations. As a result, the model choice still feels somewhat arbitrary in the current version. More broadly, the clarification suggests that the paper may need more than localized revision; it seems to require a more substantial reframing of the core story, contribution, and positioning relative to prior work.

While I appreciate the clarification, I am not sufficiently convinced that the current version can address these concerns within the scope of a short-paper revision, so my overall assessment remains unchanged.

**Summary:**

This paper presents Y-Map, a neural–symbolic framework for multi-objective ORM schema optimization. Given an object model and mapping constraints, Y-Map uses Alloy to enumerate correctness-preserving relational schema candidates. The prediction model combines lightweight symbolic schema features with a multi-encoder representation derived from pretrained models (GraphCodeBERT, CodeT5+, and LLaMA). Predicted performance vectors are used for Pareto-based selection (or optional scalarization under user-specified weights). Experiments on nine object models suggest Y-Map can achieve competitive or improved Pareto quality relative to representative baselines while reducing inference-time cost.

---

> ### Author Response · Authors · 2026-03-14
> **Response to Reviewer vaVz**
>
> We appreciate your positive assessment and your specific comments, which highlight several points that merit clearer explanation in this **short paper** version. We address them below.
>
> **1. Motivation for the multi-encoder design, including LLaMA.**
> We agree that this motivation was not adequately explained. Our intent is not to suggest that system-level metrics such as insert/query latency and memory footprint are primarily driven by LLM-style semantics. We fully agree that these metrics are mainly shaped by structural properties such as join structure, normalization/denormalization choices, inheritance strategy, and foreign-key organization.
>
> Accordingly, Y-Map does not rely on LLaMA alone or use semantic embeddings as a substitute for structural reasoning. Instead, it combines complementary views of the schema after serialization to SQL DDL: GraphCodeBERT for structural/dependency information, CodeT5+ for syntactic and schema-level regularities, and LLaMA for broader contextual abstraction. These learned representations are fused with lightweight symbolic schema features, not used in place of them.
>
> In this design, LLaMA serves as an auxiliary source of contextual signal and robustness, rather than the primary explanation of runtime behavior. We agree that this rationale should have been made more explicit.
>
>
> **2. Why this encoder set was chosen, and whether the model choice is arbitrary.**
> We agree that the paper should justify this design choice more clearly. The encoder combination was not intended as arbitrary model stacking. Rather, the goal was to avoid forcing a single representation family to capture all relevant aspects of ORM schemas equally well. Different mapping decisions can appear similar under one representation view while differing meaningfully under another. The tri-encoder design was therefore chosen to fuse structural, syntactic, and broader contextual signals, while grounding the final prediction in explicit symbolic schema features.
>
> To keep the method practical, the pretrained encoders are **frozen**, and only a lightweight regression head is trained. This substantially reduces training cost and lowers the risk of overfitting compared to end-to-end fine-tuning. In the broader extended evaluation, we also observe that the full tri-encoder configuration performs better than single-encoder and pairwise alternatives, which supports the view that these signals are complementary rather than redundant. We agree that even a brief sensitivity or ablation summary in the short-paper version would have strengthened this point.
>
> **3. Positioning relative to prior transformer-based ORM tradeoff work, such as DTS.**
> We agree that this distinction should be sharper. Y-Map’s contribution is not simply candidate enumeration with a larger model, but the combination of symbolic validity with continuous multi-objective performance prediction. For each valid schema candidate, Y-Map predicts insert latency, query latency, and memory footprint, and uses these predictions for Pareto-aware selection or optional scalarization. This differs from prior approaches focused on Pareto relevance or frontier-oriented selection, and the paper should make this distinction more explicit.
>
> **4. Clarification of the “benchmark-free” wording.**
> We agree with this comment. The phrase “benchmark-free” can be misread as implying that no measurements are collected at all. That was not our intended meaning. The intended claim is narrower: Y-Map avoids **per-candidate benchmarking at inference time**. Offline profiling is still required to construct supervision for the regression model and to compute evaluation references. We agree that the wording should be revised to make this distinction explicit and avoid overstatement.
>
> **5. Minor comments: evaluation clarity, related work, and presentation.**
> We agree with these comments. The paper should clarify the GD/HV setup more explicitly, including normalization and the hypervolume reference point. We also agree that a compact per-system breakdown or simple visualization would improve interpretability. In addition, the related-work discussion should better position the paper with respect to symbolic ORM synthesis and learning-based performance/cost modeling, and the figures should be enlarged for readability.
>
> Because this is a short paper, we prioritized the main idea, workflow, and aggregate evidence, which left limited space for these details. Your comments are therefore very helpful in identifying the clarifications that matter most. In particular, we agree that the paper should better emphasize the rationale for the multi-encoder design, the core contribution of combining symbolic validity with continuous multi-objective prediction, and the more precise inference-time meaning of the “benchmark-free” claim.

---

### Official Review · Reviewer_nxEB · 2026-03-02

**Rating:** 3
**Confidence:** 4

**Review:**

The idea of decoupling the performance evaluation and correctness is a nice idea, and novel.

However, the evaluation process is not clear.
Particularly, it is not clear exactly which dataset is used to train the regression model.
If the same set of the evaluation is used (as I understand from lines 338-340), the data leakage hinders the results.
If another set is used, data leakage should be carefully studied. Any bias in the dataset might lead to this approach outperforming the baselines on one evaluation set, and falling behind on another set.
Moreover, if the same set is used for training, the decoupling argument becomes weaker, as the user of such system still needs to deploy a large number of mappings to obtain the training data.

[Update] After clarification by the authors, my concern is addressed. The evaluation uses cross validation, so data leakage is probably minimal.

**Summary:**

This short paper presents a hybrid (neuro-symbolic) approach to ORM.
The idea is to synthesize correct mappings using the already available tool Alloy, and then instead of profiling the mappings by deploying them, which takes a lot of time and resources, predict metrics using a learned model.
The input to the metrics prediction model comes from a GNN for structural dependencies, a CodeT5+ model for syntactic and idioms, and a LLAMA model for semantics.
The results show that Y-Map outperforms two baselines both in terms of Pareto-optimality and run time.

---

> ### Author Response · Authors · 2026-03-14
> **Response to Reviewer nxEB**
>
> Thank you for your careful review and for recognizing the novelty of separating correctness-preserving synthesis from performance prediction. We appreciate your concern about the evaluation protocol and agree that it is a central issue. We also agree that the short-paper format compressed this part too much, which made the train/test protocol insufficiently clear.
>
> To clarify the key point directly: **our evaluation did not mix schemas from the same system across training and test splits.** The model was evaluated using **5-fold cross-validation at the object-model/system level**: all schemas derived from a given benchmark system were assigned entirely to either the training fold or the test fold, but never to both. This protocol was chosen specifically to reduce leakage from closely related schema variants generated from the same underlying object model. In the extended version, this is stated explicitly as: “all schemas from a system appear in either train or test, never both.”  We agree that this should have been stated much more clearly in the short-paper version as well.
>
> More concretely, the supervised dataset consists of approximately **9,500 offline-labeled schema–performance pairs** collected from **nine benchmark object models**. The short paper states that Alloy synthesis produced valid schemas across the nine models and that ground-truth labels were obtained for approximately 9,500 schemas through offline profiling.  The model is trained on these offline labels, while the pretrained encoders are frozen and only the lightweight regression head is trained.
>
> We would also like to clarify the intended meaning of the paper’s decoupling claim. Y-Map does **not** claim that no benchmarking or profiling is ever needed. Rather, the claim is that benchmarking is removed from the **online inference loop**. As stated in the short paper, the offline measurements are used to train the regressor and compute evaluation references, while **at inference time** Y-Map ranks candidates using predicted metrics without executing the workload for each candidate.  In other words, the expensive deployment-and-profile step is paid for offline once to construct supervision, and then avoided during per-candidate decision-making at inference time. We agree that this distinction should have been made more explicit.
>
> Your broader concern about **dataset bias and generalization** is also well taken. Any learned predictor depends on the representativeness of the offline-labeled corpus, and its accuracy may degrade under substantial workload shift, DBMS changes, or different deployment conditions. The short paper already acknowledges this limitation, noting that accuracy depends on workload representativeness and that large shifts in workload may require relabeling or fine-tuning.  We agree that this threat to validity warrants greater emphasis in the evaluation discussion.
>
> We also appreciate your suggestion that training and evaluation should be separated as much as possible. That was exactly the intent of our object-model-level cross-validation protocol: to avoid evaluating on arbitrarily mixed schemas from the same source system and instead test on held-out systems within each fold. While the current short-paper wording may have suggested otherwise, the actual protocol was designed to address this leakage concern precisely.
>
> Because this submission is a **short paper**, several implementation and protocol details were compressed in favor of the main idea and aggregate results. Your comment makes clear that the evaluation protocol itself needed more explicit presentation, and we appreciate that observation. In a revision, we would clarify:
> (1) that the split is performed at the system/object-model level,
> (2) that the 9,500 labels come from offline profiling only, and
> (3) that “benchmark-free” refers specifically to **inference-time optimization**, not the absence of offline supervision.
>
> Thank you again for highlighting this issue. We appreciate the opportunity to clarify it more precisely.

---

> > ### Comment · Reviewer_nxEB · 2026-03-14
> >
> > Thank you for the clarification. I have updated my review accordingly.
> > Please make sure to clarify the evaluation process in the paper as well.

---

> > > ### Author Response · Authors · 2026-03-14
> > > **Response to Reviewer nxEB**
> > >
> > > Thank you for the follow-up and for updating your review. We appreciate your careful consideration and constructive feedback. We agree that the evaluation protocol should be stated more explicitly in the paper, and we will clarify it in the revised version.

---

### Official Review · Reviewer_4FLm · 2026-03-11

**Rating:** 3
**Confidence:** 2

**Review:**

I am not an expert on database systems and the related literature on ORM optimization work. At a high-level the core idea seems novel, which uses Alloy to synthesize ORM schemas that adheres to specified constraints, then trains a model to predict continuous vectors of performance metrics. Following a user-specified objective, there is a selection that optimizes for the provided objective.

For a short-paper the presented concepts and evaluation is reasonable and intriguing to find the results are promising as well. However, I suggest the authors address the following comment and make the paper stronger.

* There is a mismatch in what authors claim in the introduction and content, benchmark-free attribute of this work. Although there is a discussion to clarify, I would avoid using such claims for the paper as it misleads the reader.
* Why use 3 different embeddings and not create one embedding that captures all necessary properties? What is the trade-off of this design choice can be explained in bit more details
* What is the effort to train this model and how many data points are used to train the model for the evaluation? Some details can help the reader.
* A threat to validity is the accuracy of the offline-labeled dataset that should be discussed.


Minor Comment:
- Table-1 captures results of 9 systems, however the first paragraph in Section 5.2 refers to 11 systems?

**Summary:**

This paper presents a neural-symbolic framework named Y-map, that synthesizes ORM schemas which are correct by construction, and ranks candidate schemas using a multi-objective regression model that predicts a continuous vector capturing various performance metrics. Evaluation across nine object models, the authors show Y-Map outperforms selected baselines.

---

> ### Author Response · Authors · 2026-03-14
> **Response to Reviewer 4FLm**
>
> Thank you for your thoughtful and constructive review. We sincerely appreciate your positive assessment of the paper’s core idea and your recognition that, for a short paper, the proposed concepts and initial evaluation are promising. We also appreciate the specific points you raised. They are helpful, and we agree that several aspects would benefit from clearer wording and additional clarification.
>
> First, regarding the term **“benchmark-free,”** we agree that it can be misinterpreted when stated without qualification. Our intended claim is specifically **benchmark-free at inference time**, not that no measurements are ever collected during the overall pipeline. Y-Map still relies on offline profiling to construct supervised training data and evaluation references. However, once the model is trained, it can rank synthesized valid ORM schemas **without deploying each candidate, loading data, or executing the workload for every candidate at inference time**. We agree that this distinction should be stated more explicitly. In a revision, we would replace or qualify the term to avoid misleading readers.
>
> Second, regarding the use of **three different embeddings**, our design goal was to capture **complementary views** of a schema rather than rely on a single representation to encode all relevant information. In our framework, the different encoders contribute different signals: structural relationships, syntactic/schema-level regularities, and broader semantic or contextual patterns from serialized schema representations. We did not assume that a single embedding model would reliably capture all of these aspects equally well, especially given the complexity of ORM design tradeoffs. The tradeoff of this choice is a somewhat heavier representation pipeline, but the benefit is a richer feature space for predicting continuous performance metrics. To control training cost and reduce overfitting risk, the pretrained encoders are kept frozen, and only a lightweight regression head is trained. We agree that this rationale should be explained more clearly in the paper.
>
> Third, regarding **training effort and dataset size**, we agree that readers would benefit from more explicit information. The evaluation uses approximately **9,500 offline-labeled schema–performance pairs** derived from **nine benchmark object models**. The model is evaluated using **cross-validation**, and the encoders remain frozen while only the regression head is trained. This design keeps training significantly lighter than end-to-end fine-tuning of large pretrained models. We will make these details more explicit so that the practical cost of training is easier to understand.
>
> Fourth, we agree that the **quality and representativeness of the offline-labeled dataset** is an important **threat to validity** and deserves more explicit discussion. The predictive model depends on the quality of the offline measurements used as supervision, and its usefulness may be affected by workload shift, DBMS configuration changes, or other environmental differences between labeling and deployment conditions. Our intent is not to claim that offline labeling is perfect or universally transferable, but rather to move the expensive per-candidate benchmarking cost out of the online decision loop. We agree that the paper should state this limitation more directly.
>
> Finally, thank you for pointing out the inconsistency between **Table 1**, which reports results over **9 systems**, and the text in **Section 5.2**, which refers to **11 systems**. You are correct that this is confusing. The short-paper results summarized in the table correspond to the **nine benchmark systems** used for the main evaluation, while the reference to eleven systems reflects wording carried over from the broader extended evaluation. We agree that this should be corrected for consistency.
>
> More broadly, we also note that this submission was prepared as a **short paper**, so some methodological and evaluation details were necessarily compressed. That said, we appreciate your feedback because it helps us identify which clarifications are most important to readers, and we will make those points more explicit.
>
> Thank you again for the encouraging and constructive review.

---

### Author Response · Authors · 2026-03-14
**Global Response to Reviewers**

We thank all reviewers for their thoughtful and constructive feedback. We appreciate the recognition that the problem is important, that the overall pipeline is clear, and that the results are promising. As this submission was prepared as a **short paper**, several methodological and positioning details were necessarily compressed. The discussion has been very helpful in identifying which clarifications are most important. We summarize them below.

**1. Clarification of the evaluation protocol and leakage concern.**
We agree that the evaluation process should have been stated more explicitly. The model was evaluated using **5-fold cross-validation at the object-model/system level**: all schemas derived from a given benchmark system were assigned entirely to either the training or test split within each fold, but never to both. This design was intended specifically to reduce leakage from closely related schema variants. The supervised dataset contains approximately **9,500 offline-labeled schema–performance pairs** collected from **nine benchmark object models**.

**2. Clarification of the “benchmark-free” claim.**
We agree that the term **“benchmark-free”** can be misread if left unqualified. Our intended claim is specifically **benchmark-free at inference time**. Y-Map still uses offline profiling to construct supervision for the regression model and to compute evaluation references. However, once trained, Y-Map can rank synthesized valid ORM schemas **without deploying each candidate, loading data, or executing the workload for every candidate at inference time**. We agree that the paper's wording should be more precise.

**3. Motivation for the multi-encoder design.**
We agree that the rationale for the tri-encoder design should be clearer. Our intent is not to claim that system-level metrics are primarily driven by LLM-style semantics. Rather, Y-Map combines **complementary views** of the schema: structural/dependency information, syntactic/schema-level regularities, broader contextual abstraction from serialized SQL DDL, and lightweight symbolic schema features. In this design, the LLM-derived representation serves as an **auxiliary source of contextual signal and robustness**, not as the primary explanation of runtime behavior. The pretrained encoders are frozen, and only a lightweight regression head is trained.

**4. Positioning relative to prior work.**
We agree that the distinction from prior transformer-assisted ORM tradeoff methods should be made more explicit. Y-Map’s contribution is not simply candidate enumeration with a larger model, but the combination of **symbolic validity** with **continuous multi-objective performance prediction**. For each valid schema candidate, Y-Map predicts a continuous three-objective vector—insert latency, query latency, and memory footprint—and uses those predictions for **Pareto-aware selection** or optional scalarization. This differs from approaches centered on Pareto relevance or frontier-oriented selection, and we agree that the paper should make this distinction more explicit.

**5. Threats to validity and presentation details.**
We also agree that the paper should more explicitly discuss the limitations of the offline-labeled dataset, including its dependence on workload representativeness and possible degradation under workload or environmental shifts. In addition, the paper should better clarify the GD/HV setup, improve figure readability, and more clearly situate the work relative to symbolic ORM synthesis and learning-based performance/cost modeling.

**6. Clarification on the 9 vs. 11 systems wording.**
Thank you for catching this inconsistency. The aggregate results reported in the short paper correspond to **nine benchmark systems**. The mention of **eleven systems** reflects wording carried over from the broader extended evaluation and should be corrected for consistency.

We sincerely appreciate the reviewers’ feedback. It helped identify the main points that require clearer explanation in this **short-paper** version, and we will make these clarifications more explicit.